# Expression of most retrotransposons in human blood correlates with biological aging

Yi-Ting Tsai[1†], Nogayhan Seymen[1†], I Richard Thompson[1], Xinchen Zou[2], Warisha Mumtaz[1], Sila Gerlevik[1], Ghulam J Mufti[1], Mohammad M Karimi[1]*

[1]Comprehensive Cancer Centre, School of Cancer & Pharmaceutical Sciences, Faculty of Life Sciences & Medicine, King's College London, London, United Kingdom; [2]MRC LMS, Imperial College London, London, United Kingdom

## eLife Assessment

The study by Tsai et al. employed multi-omics approaches, including transcriptomic, methylomic, and single-cell RNA-seq, and provided a **solid** and comprehensive analysis of the correlation between retrotransposable element (RTE) expression and biological aging in human blood. Their findings highlight the differential roles of RTE families, providing **valuable** insights for understanding the mechanisms of human aging.

**\*For correspondence:**
mohammad.karimi@kcl.ac.uk

†These authors contributed equally to this work

**Abstract** Retrotransposons (RTEs) have been postulated to reactivate with age and contribute to aging through activated innate immune response and inflammation. Here, we analyzed the relationship between RTE expression and aging using published transcriptomic and methylomic datasets of human blood. Despite no observed correlation between RTE activity and chronological age, the expression of most RTE classes and families except short interspersed nuclear elements (SINEs) correlated with biological age-associated gene signature scores. Strikingly, we found that the expression of SINEs was linked to upregulated DNA repair pathways in multiple cohorts. We also observed DNA hypomethylation with aging and the significant increase in RTE expression level in hypomethylated RTEs except for SINEs. Additionally, our single-cell transcriptomic analysis suggested a role for plasma cells in aging mediated by RTEs. Altogether, our multi-omics analysis of large human cohorts highlights the role of RTEs in biological aging and suggests possible mechanisms and cell populations for future investigations.

## Introduction

Transposable elements (TEs) are genetic elements that can move within the genome and are categorized into DNA transposons and RTEs, which depend on cDNA intermediates to function. RTEs include three classes: endogenous retrovirus long terminal repeats (LTRs), and long and short interspersed nuclear elements, known as LINEs and SINEs, respectively. LINEs and SINEs employ target-primed reverse transcription for genome integration, with SINEs depending on the proteins encoded by LINEs (*Bourque et al., 2018*). Although most of the RTE sequences are dormant in the host genome, they continue to play vital roles in human evolution and physiology (*Gorbunova et al., 2014*; *Gorbunova et al., 2021*).

RTEs are silenced through heterochromatinization and DNA methylation in the early developmental stages as part of the host defense mechanism (*Gorbunova et al., 2021*; *Van Meter et al., 2014*). However, increased RTE activity that occurs with aging can lead to genome instability and

activation of DNA damage pathways (*Van Meter et al., 2014*; *Tsurumi and Li, 2012*; *De Cecco et al., 2013b*; *Wood et al., 2016*). Additionally, accumulation of cytoplasmic RTE cDNAs detected in aging organisms and senescent cells can activate type I interferon (IFN-I) response and inflammation (*Gorbunova et al., 2021*; *Van Meter et al., 2014*; *De Cecco et al., 2019*; *Simon et al., 2019*; *De Cecco et al., 2013a*), that contributes to inflammaging (*Franceschi et al., 2000*). Furthermore, chromatin remodeling and cellular senescence also contribute partially to the RTE reactivation (*De Cecco et al., 2013a*). Cellular senescence is a non-proliferative state elicited by stress factors such as DNA damage and is closely intertwined with inflammation and aging (*Gorgoulis et al., 2019*). Through the expression of senescence-associated secretory phenotype (SASP), senescent cells promote inflammation and senescence in other cells, including immune cells, leading to a compromised immune system and a vicious cycle of inflammation (*Coppé et al., 2010*). Understanding the relationship between RTE expression and SASP, cellular senescence, and inflammaging could offer important insights into strategies against aging and age-related diseases (ARDs).

Recent studies have documented the relationship between RTE activation and biological age-related (BAR) events, but comprehensive and large-scale studies are lacking, mainly due to the paucity of RNA-seq data for large non-cancerous human cohorts. Most of the existing transcriptomics datasets are microarray-based, where the computational methods to analyze repetitive elements have not been sufficiently developed. By overlapping Illumina microarray expression and methylation probe locations to RTE locations in RepeatMasker (*Smit et al., 2013*), we were able to identify a sufficient number of probes to calculate the expression and methylation levels of RTE classes and families.

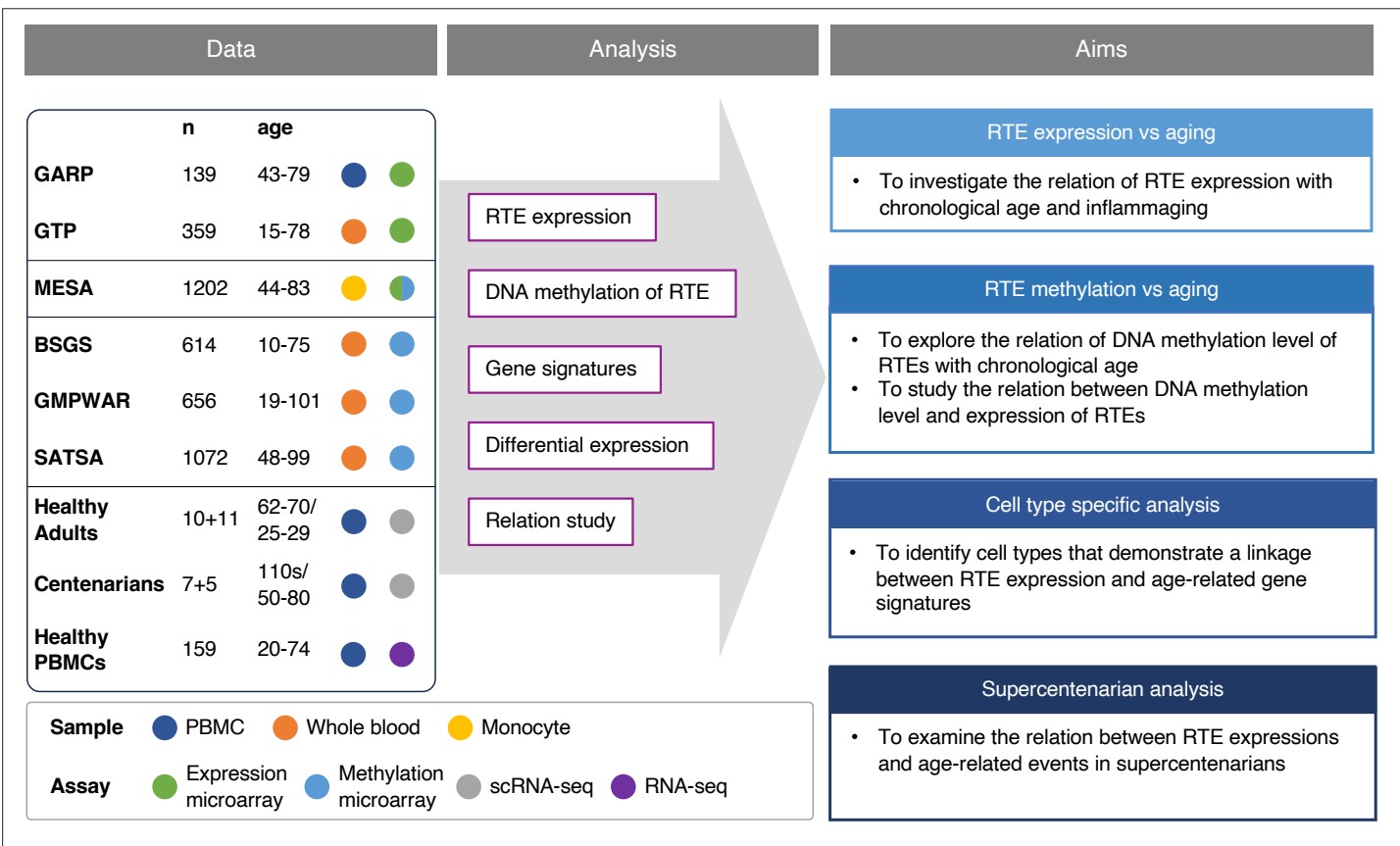

**Figure 1.** Conceptual framework and the study design. We collected published datasets of human blood samples for gene expression, DNA methylation, and single-cell transcriptomic data. The analysis aimed to study the relation between the expression and DNA methylation of retrotransposons (RTEs) versus chronological and biological aging in large human cohorts. The single-cell transcriptomic datasets were employed for cell type-specific analysis of RTEs in peripheral blood mononuclear cell (PBMC) to identify the relation between RTE expression and aging events for annotated cell types within old versus young PBMC samples.

The online version of this article includes the following figure supplement(s) for figure 1:

**Figure supplement 1.** Number of retrotransposon (RTE)-covering probes.

Building on this methodology, we explored how RTE expression contributes to biological aging using publicly available transcriptomics microarray data derived from human blood samples. More specifically, we first investigated if RTE expression was correlated with chronological age, and then we analyzed the relationship between RTE and BAR events including cellular senescence, inflammation, and IFN-I response with published gene signatures. Using microarray methylomic data, we also investigated the DNA methylation level of RTEs in blood samples of multiple non-cancerous human cohorts and examined the relation between DNA methylation and RTE expression and aging. Furthermore, with annotated single-cell transcriptomic data of the peripheral blood mononuclear cells (PBMCs) of 21 healthy human samples from a recent study of aging immunity (*Mogilenko et al., 2021*), we identified cells that could be implicated in the process of RTE reactivation and aging. We also validated our microarray result with a publicly available bulk RNA-seq PBMC data from healthy individuals that was recently published in a new study of aging (*Morandini et al., 2024*). Lastly, the single-cell transcriptomic data of supercentenarians *Hashimoto et al., 2019* was analyzed to examine the interplay between RTE expression and aging as a factor of aging and longevity (*Figure 1*).

## Results

### Analyzing DNA methylation and expression levels of RTEs using microarray data

We collected three published microarray datasets from large-scale human studies, including the peripheral monocyte samples from the Multi-Ethnic Study of Atherosclerosis (*Bild et al., 2002*) (MESA, aged 44–83, n=1202), the whole blood (WB) samples from Grady Trauma Project (*Binder et al., 2008*; *Gillespie et al., 2009*) (GTP, aged 15–77, n=359), and PBMC samples from Genetics, Osteoarthritis and Progression (*Riyazi et al., 2005*) (GARP, aged 43–79, n=139), which are all non-cancerous human samples (*Supplementary file 1*). The studies were conducted using either Illumina HumanHT-12 V3 or Illumina HumanHT-12 V4 expression microarray kits. After comparing the probes of the two microarray versions, we adopted the more comprehensive probe list of V4, which contains the full intersection of the two lists. To quantify RTE expression, we mapped the microarray probe locations to RTE locations in RepeatMasker to extract the list of noncoding (intergenic or intronic) probes that cover the RTE regions. We included three main RTE classes: (1) the LINE class, which encompasses the L1 and L2 families; (2) the SINE class, with Alu and MIR as the two main families; and (3) the LTR class which comprises the ERV1, ERVL, ERVL-MaLR, and ERVK families. Most of the RTE-covering probes available on Illumina HumanHT-12 V4 are present in MESA and GARP, while fewer are available in GTP (*Figure 1—figure supplement 1a*, *Supplementary files 2–4*).

Four methylation datasets were analysed, including MESA, Swedish Adoption/Twin Study of Aging (*Wang et al., 2018*) (SATSA, aged 48–98, n=1072), Brisbane Systems Genetics Study (*Powell et al., 2012*) (BSGS, aged 10–75, n=862), and Genome-wide Methylation Profiles Reveal Quantitative Views of Human Aging Rates (*Hannum et al., 2013*) (GMPWAR, aged 19–101, n=656). SATSA, BSGS, and GMPWAR include genome-wide DNA methylation of WB samples produced by Illumina Infinium 450 k array (*Supplementary file 1*). The DNA methylation probes from the Illumina Infinium 450 k array kit were aligned to the locations in RepeatMasker to identify the probes overlapping the RTE regions (*Supplementary file 5*). More than 90% of the RTE-covering probes are present in MESA and GMPWAR, while fewer, but more than half, are available in SATA and BSGS datasets (*Figure 1—figure supplement 1b*).

### The chronological age is not linked with RTE expression

Firstly, we examined the relationship between RTE expression and chronological age in the MESA, GARP, and GTP cohorts. The ages of the individuals enrolled in these studies range from 15 to 83 years, with the mean being 70.2 (MESA), 60.1 (GAPR), and 42.5 (GTP) (*Supplementary file 1*). Strikingly, no significant relationship (*p-value* < 0.05) is found between chronological age and expressions of the three RTE classes across the datasets (*Figure 2a*). Similarly, apart from a weak correlation observed between a few RTE families and chronological age, we could not identify any strong relation between chronological age and expression of RTE families across the three cohorts (*Figure 2—figure supplement 1a*).

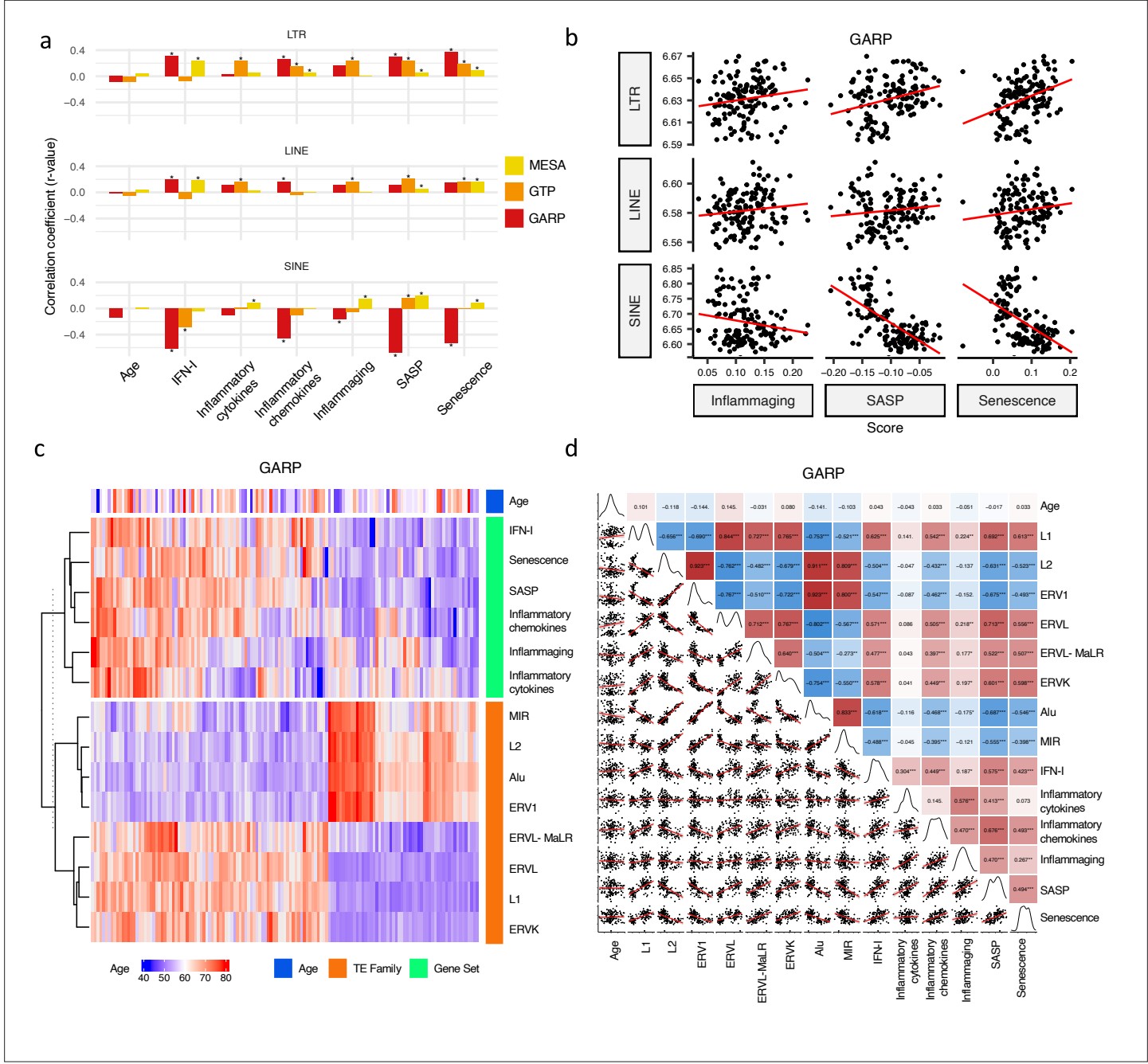

**Figure 2.** Correlation analysis between retrotransposon (RTE) expression, chronological age, and age-associated gene signature scores. (**a**) No correlation between RTE expression and chronological age versus positive correlations between biological age-related (BAR) gene signature scores and LINE and long terminal repeat (LTR) expressions. Pair-wise correlation coefficients were calculated between the expression of different RTE classes (LTR, LINE, and SINE) and chronological age and six BAR gene signature scores in monocytes (Multi-Ethnic Study of Atherosclerosis, MESA), peripheral blood mononuclear cells (PBMCs) (GARP), and the whole blood (WB) (GTP). (**b**), Scatter plots displaying a positive correlation between LINE and LTR expressions and inflammaging, senescence associated secretory phenotype (SASP), and senescence gene signature scores in PBMCs. (**c**), Different families of RTEs were divided into two major groups based on their correlation and inverse correlation with BAR gene signature scores in PBMC samples. (**d**), Correlation matrix depicting all pair-wise combinations to identify the correlation between chronological age, RTE family expressions, and six age-associated signature scores in PBMCs. **p≤0.01, ***p≤0.001, Pearson's correlation. MESA, n=1202; GARP, n=139; GTP, n=359.

The online version of this article includes the following figure supplement(s) for figure 2:

**Figure supplement 1.** Correlation analysis between biological age-related (BAR) gene signatures and retrotransposon (RTE) family expressions in human cohorts.

*Figure 2 continued on next page*

*Figure 2 continued*

**Figure supplement 2.** Comparing SINE expression in 25 cell types obtained from young versus old peripheral blood mononuclear cell (PBMC) human samples.

**Figure supplement 3.** Comparing LINE expression in 25 cell types obtained from young versus old peripheral blood mononuclear cell (PBMC) human samples.

**Figure supplement 4.** Comparing long terminal repeats (LTR) expressions in 25 cell types obtained from young versus old peripheral blood mononuclear cell (PBMC) human samples.

**Figure supplement 5.** Correlation analysis of retrotransposon (RTE) expression and chronological age in RNA-seq data of healthy human peripheral blood mononuclear cell (PBMC) samples.

**Figure supplement 6.** Correlation analysis of retrotransposon (RTE) expression and type I interferon (IFN-I) score in RNA-seq data of healthy human peripheral blood mononuclear cell (PBMC) samples.

**Figure supplement 7.** Increasing trend of biological age-related (BAR) gene signature scores in high vs low long terminal repeats (LTR) and LINE expression groups in the three human cohorts.

**Figure supplement 8.** Different patterns of biological age-related (BAR) signature scores in Multi-Ethnic Study of Atherosclerosis (MESA) vs GARP vs Grady Trauma Project (GTP) for high vs low SINE and Alu expression groups in the three human cohorts.

A similar correlation analysis was carried out in a scRNA-seq data of the PBMCs of 21 non-obese healthy men annotated into 25 cell types by *Mogilenko et al., 2021* (*Supplementary file 6*). When comparing the RTE expression of the young (aged 25–29, n=11) group with the old (aged 62–70, n=10) group, we could not find any significant difference or trend in any cell types between the two groups (*Figure 2—figure supplements 2–4*).

To investigate whether our observation is consistent in whole transcriptomic data, we conducted correlation analysis on a set of RNA-seq data on healthy human PBMC (*Morandini et al., 2024*) (aged 20–74, n=159). We did not observe any statistically significant correlation between chronological age and expression of RTE classes or families (*Figure 2—figure supplement 5*). Overall, RTE expression did not correlate with chronological aging across the microarray, scRNA-seq, and RNA-seq datasets.

## RTE expression positively correlates with BAR gene signature scores except for SINEs

In mammals, aging is characterized by an intricate network of inflammation, IFN-I signaling, and cellular senescence (*Gorbunova et al., 2021*). To measure the level of biological aging from the transcriptomic data, we focused on six gene sets retrieved from widely cited studies: IFN-I (*Ivashkiv and Donlin, 2014*), inflammatory cytokines (*Mogilenko et al., 2022*), inflammatory chemokines (*Danaher et al., 2017*), inflammaging (*Franceschi and Campisi, 2014*), SASP (*Hudgins et al., 2018*), and cellular senescence (*Troiani et al., 2022*; *Supplementary file 7*). The scores of these six age-associated gene sets were calculated for each individual in the MESA, GTP, and GARP cohorts by applying the *singscore* package in Bioconductor (*Foroutan et al., 2018*). Except for GTP, in which 12 genes were missing from the gene sets in total (*Supplementary file 8*), MESA and GARP microarray datasets had the expression of all the genes in the gene sets.

We analyzed the relation between RTE expression and the six BAR gene signature scores. Positive correlations were observed between the expressions of LINEs and LTRs and BAR gene signature scores across three datasets, while inverse correlation was present between the expression of SINEs and these scores in the PBMC samples from the GARP cohort (*Figure 2a and b*). The discrepant pattern of LTRs and LINEs versus SINEs in PBMC is more pronounced in the correlation analysis between the expression of RTE families and the BAR gene sets (*Figure 2c and d*). Interestingly, the expressive patterns of RTE families are divided into two groups that also reflect their positive or inverse correlations with BAR signature scores in the GARP cohort. In the GARP cohort, the strongest positive correlation with BAR scores was observed in L1, ERVK, and ERVL. By contrast, the expressions of Alu and MIR of SINE, L2, and ERV1 were inversely correlated with BAR signature scores in the GARP cohort (*Figure 2c and d*). This pattern was also observed in the MESA peripheral monocyte samples for L1, ERVL, and ERVK, but not other RTE families (*Figure 2—figure supplement 1a and c*). The WB GTP cohort showed divergent results to the other two cohorts perhaps because of the absence of certain genes in the six BAR gene sets or a lower number of TE probes in GTP (*Figure 1—figure supplement 1b*, *Figure 2a–b* and *Supplementary file 8*).

To study the association between RTE expression and BAR gene signatures, we also split the samples in each cohort based on quartiles of RTE expression into high, low, and medium (top 25%, bottom 25%, and middle 50%) groups and compared the BAR gene signature scores across these three groups. Across the datasets, there is an overall trend of increased scores of BAR gene signatures in high versus low LINE and LTR expression groups (*Figure 2—figure supplement 7*). In fact, amongst all gene sets, we found the most significant increase for SASP and senescence and to a lesser extent for inflammaging signatures. We observed higher SASP, inflammatory cytokine, senescence, and inflammaging in the peripheral monocytes (MESA) in high SINE- and Alu-expressing groups (*Figure 2—figure supplement 8*). However, in the PBMC samples of the GARP cohorts, groups of high SINE and Alu expressions demonstrate lower expressions of SASP, inflammatory chemokine, senescence, and inflammaging, that reflects the inverse correlation we observed before (*Figure 2c*). Interestingly, when we examined RNA-seq PBMC data, we found that there is a significant correlation between IFN-I signature score and expression of LINE and LTR classes and most of their families (*p-value* <0.01), but such a significant correlation did not exist between IFN-I score and SINE class and Alu family (*Figure 2—figure supplement 6*). There are also no specific trends for SINE expression in the WB samples from the GTP cohort (*Figure 2—figure supplement 6*). Taken together, SINEs display different patterns than LINEs and LTRs in terms of the association between their expressions and BAR gene signatures. This prompted us to conduct a gene set variation analysis (GSVA) *Hänzelmann et al., 2013* to identify the pathways that are regulated in the high versus low RTE expression groups in the three cohorts.

## Elevated SINE expression is linked with the upregulation of the DNA repair pathways, while elevated LINE and LTR expressions are associated with the inflammatory response

We conducted GSVA for high versus low expression groups of RTE classes and families to identify the pathways that might be affected by RTE expression. We specifically focused on the pathways (gene sets) related to the inflammatory and DNA repair responses retrieved from the Molecular Signatures Database (MSigDB) (*Subramanian et al., 2005*) with 'inflammatory' and 'dna_repair' as search keywords.

We found that a significant fraction of gene sets related to DNA repair were downregulated in the group of monocyte samples (MESA cohort) highly expressing LINE and LTR classes but upregulated in the monocyte and PBMC samples (MESA and GARP cohorts, respectively) highly expressing SINEs (*Figure 3a*). At the family level, there is a unique increase in DNA repair activity in the monocytes of the cohort with high Alu expression. In the PBMC samples of the GARP dataset, DNA repair pathways are upregulated in groups highly expressing L2, ERV1, Alu, and MIR, while most of the other RTE classes demonstrate decreased DNA repair activity. In contrast, the inflammatory response is upregulated in the monocyte and PBMC samples of groups expressing LINE and LTR classes and families at high levels, and such upregulation is less common in SINE (*Figure 3a*). These observations provide insight into the perpetuating contrast between SINE versus LINE and LTR observed in our analysis, as SINE might contribute to aging via genome instability instead of inflammation that is associated with LINE and LTR activities. Overall, our observations are consistent across MESA and GARP cohorts but less so in the GTP cohort (*Figure 3b*), which might be attributed to less RTE-covering probes and the missing genes from the BAR gene signatures. (*Figure 1—figure supplement 1*, *Supplementary file 8*).

## DNA methylation levels of RTEs are inversely correlated with the chronological age and the RTE expression except for SINE expression

The demethylation of RTEs is suggested to contribute to aging (*Gorbunova et al., 2021*; *Tsurumi and Li, 2012*; *Wood et al., 2016*). To examine the effect of RTE demethylation on chronological aging, we analyzed DNA methylation data from multiple cohorts including monocytes from the MESA cohort and the WB data from GMPWAR, BSGS, and SATSA. Our analysis showed an inverse correlation between the methylation level of RTEs and chronological age across all RTE classes and families in the WB and monocytes (*Figure 4a–b* and *Figure 4—figure supplement 1*). This observed trend is in line with previous reports of demethylation of RTE with age in gene-poor regions (*Tsurumi and Li, 2012*; *Wang et al., 2018*; *Hannum et al., 2013*).

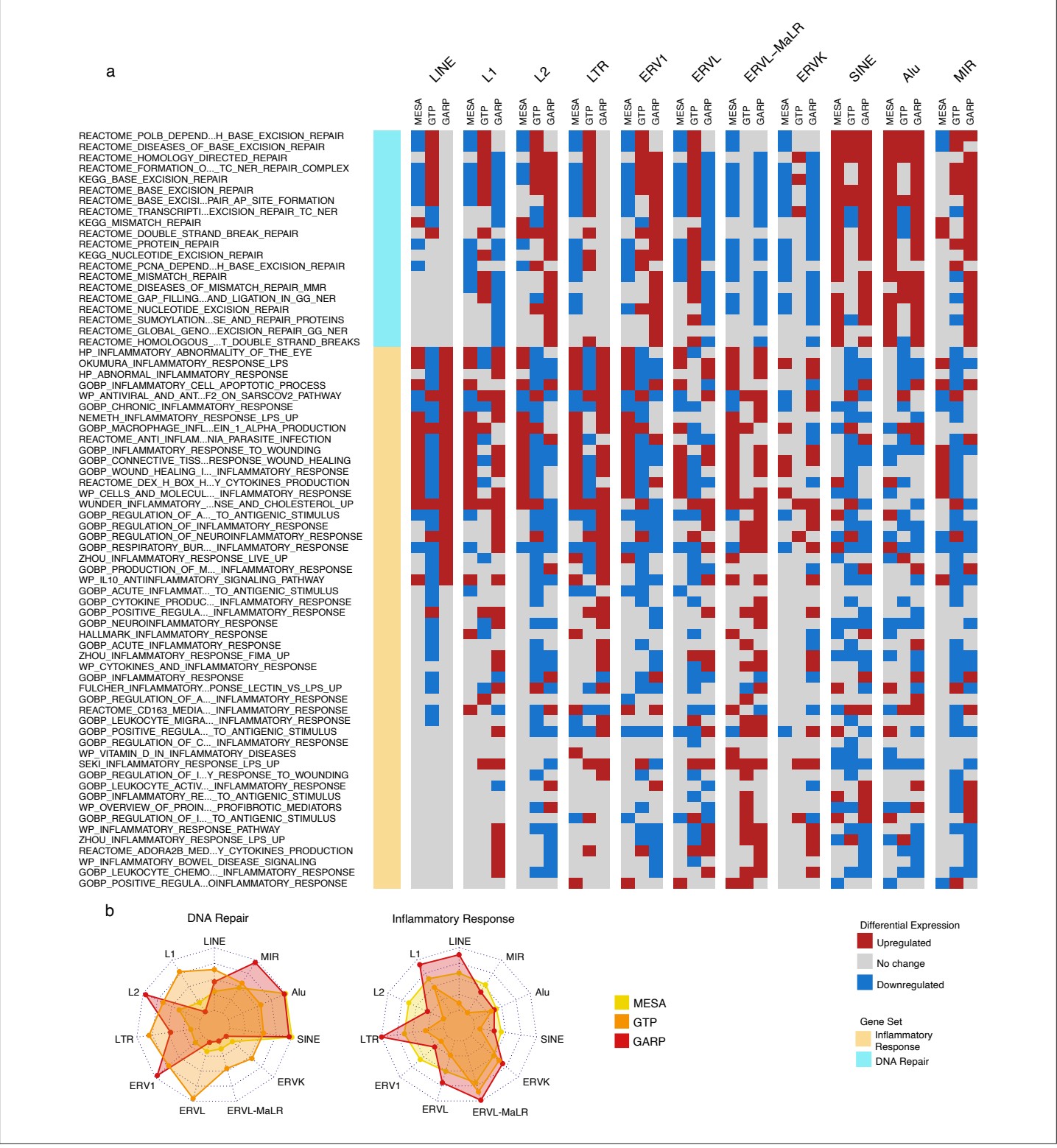

**Figure 3.** Upregulation of DNA repair vs inflammatory responses for samples with high expression of SINE vs long terminal repeats (LTR) and LINE in the Multi-Ethnic Study of Atherosclerosis (MESA) and GARP cohorts. (**a**) Gene set variation analysis (GSVA) demonstrates the increased activity of DNA repair pathways in the group of samples with high vs low SINE expression in the MESA and GARP cohorts. In contrast, the inflammatory response is upregulated in the sample groups highly expressing LINE and LTR classes and families in the MESA and GARP cohorts. The samples in each cohort were divided into low (first quartile), medium (second and third quartile), and high (fourth quartile) expression groups based on the expression of

*Figure 3 continued on next page*

*Figure 3 continued*

retrotransposon (RTE) classes or families. GSVA was applied on high vs low groups for each class and family of RTEs. The threshold for differential expression is set at |logFC|>0.1 and p<0.05. (**b**), The Radar plot shows the difference between the number of upregulated versus downregulated gene sets related to DNA repair and inflammatory response in each cohort. While high expression of SINE and Alu is significantly associated with a high number of up-regulated DNA-repair gene sets, LINE and L1 expressions are highly related to the high number of activated gene sets related to inflammatory response in the MESA and GARP cohorts. This result is not highly supported by the GTP cohort, more likely due to the low number of probes in this cohort.

To elucidate the relationship between the expression and methylation level of RTEs, we divided the MESA samples, with matched RTE DNA methylation and expression, into three groups based on the level of expression of RTEs in different classes and families: low (first quartile), medium (second and third quartile), and high (fourth quartile) expression groups of monocyte samples from the MESA cohort. The DNA methylation levels of LINEs and LTRs significantly decreased in groups of high LINE and LTR expressions, respectively, but SINEs and the satellite repeats (as negative control) did not display the same pattern (*Figure 4c*). LINE families, MIR, and LTR families except ERVK displayed lower levels of methylation in higher expression groups, while such patterns was absent in Alu, CR1, and ERVl (*Figure 4—figure supplement 2*). Although the DNA methylation level of SINEs positively correlated with the methylation levels of LINEs and LTRs, the expression of SINEs was not significantly correlated with the expression levels of other RTE classes (*Figure 4d*). Overall, we found a distinct pattern for SINEs versus LTRs and LINEs by exploring a relationship between DNA methylation and expression levels of RTEs in monocytes, consistent with our other findings showing different patterns for SINEs versus LTRs and LINEs in PBMCs (see above). This result also suggests a dispensable role of DNA methylation for SINE silencing in monocytes, which is in line with previous studies showing SINEs are not derepressed by deletion of DNA methyltransferases or treatment with DNA demethylating agents and are primarily regulated by histone modifications (*Varshney et al., 2015*).

## Elevated level of RTEs in plasma cells of healthy PBMC samples is associated with the high SASP and inflammaging gene signature scores

As described above, we did not identify any association between RTE expression and chronological aging for any human cohorts we analyzed. However, the positive correlation between LINE and LTR expressions with BAR gene signature scores in the PBMC samples of the GARP cohort prompted us to investigate PBMC samples in more detail using PBMC scRNAseq data of 21 samples annotated into 25 cell types by *Mogilenko et al., 2021* (*Supplementary file 6*). The scRNA-seq samples were divided based on low and high BAR gene signature scores to compare against the expression of RTE classes for each cell type. Among the 25 annotated cell types, plasma cells consistently demonstrate significantly elevated RTE expression in cells of high SASP expression that is not observed in the other cell types (*Figure 5a* and *Figure 5—figure supplements 1–3*). Increased LINE and SINE expressions were also observed in the plasma cells of the samples with high inflammatory chemokines and inflammaging gene signature scores (*Figure 5b and c*). Although not statistically significant, elevated RTE expression was also seen in the groups with high inflammatory cytokines and senescence but not IFN-I scores (*Figure 5d–f*). Overall, this finding indicates that plasma cells might play an important role in the process in which reactivation of RTEs regulates aging.

## Downregulation of RTE expression in supercentenarians versus normal-aged cases despite high BAR signature scores in NK and T cells of supercentenarians

Supercentenarians who have reached 110 years of age are an excellent resource for investigating healthy aging. We hypothesized that supercentenarians should express RTEs less than normal-aged people because RTE expression, particularly LINE and LTR expressions, can activate the inflammatory pathways (*Figure 3*), which would be generally harmful to healthy aging. To understand whether there is any RTE expression change in supercentenarians versus normal-aged people and identify the relationship to BAR gene signatures, we compared the scRNA-seq data of the PBMCs of seven supercentenarians (over 110 years) to five non-centenarian (50–80 years) controls (*Hashimoto et al., 2019*). We collected the annotated cells of these twelve cases as a Seurat object for which the cells were annotated into ten different cell types. Compared with the control group, the supercentenarians

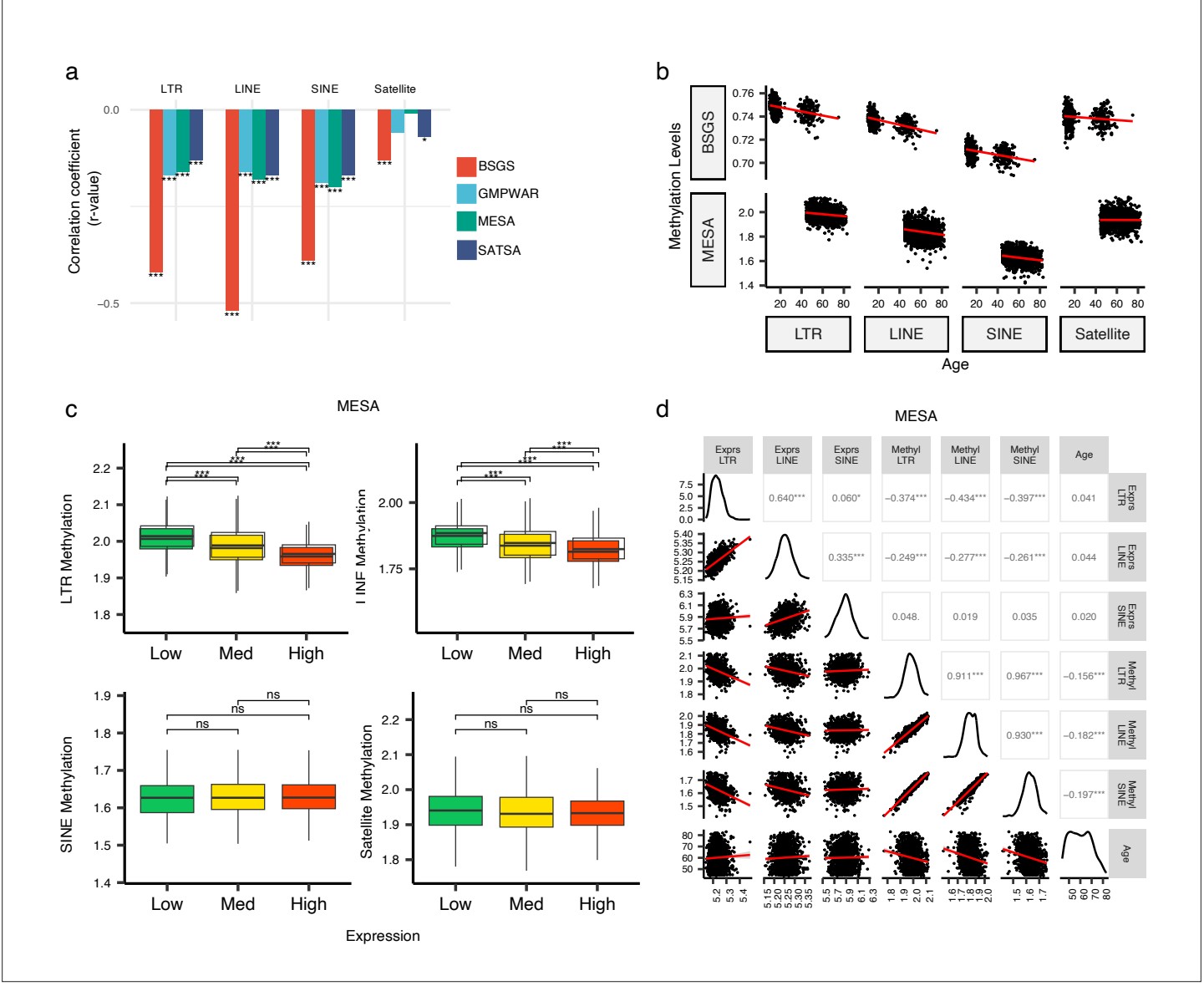

**Figure 4.** Inverse correlation of DNA methylation levels of retrotransposons (RTEs) with the chronological age and the RTE expression except for SINE expression. (**a**, **b**), Methylation levels of RTE classes inversely correlated with chronological age in monocyte (Multi-Ethnic Study of Atherosclerosis, MESA) and whole blood (WB) (BSGS, SATSA, and GMPWAR) samples. Satellite DNA was included as a control group. (**c**), Methylation levels versus low (first quartile), medium (second and third quartile), and high (fourth quartile) expressions of RTE classes in monocytes (MESA). Wilcoxon test; ns: not significant. (**d**), Correlation matrix for RTE expressions and methylation levels, and chronological age. *p≤0.05, **p≤0.01, ***p≤0.001, ****p≤0.0001, Pearson's correlation. MESA, n=1202; BSGS, n=614; GMPWAR, n=656; SATSA, n=1072.

The online version of this article includes the following figure supplement(s) for figure 4:

**Figure supplement 1.** Correlation of methylation levels of retrotransposon (RTE) families with chronological age in monocyte (Multi-Ethnic Study of Atherosclerosis, MESA) and whole blood (WB) (BSGS, SATSA, and GMPWAR) samples.

**Figure supplement 2.** Methylation levels versus low (first quartile), medium (second and third quartile), and high (fourth quartile) expressions of retrotransposon (RTE) families in monocytes.

demonstrated decreased RTE expression in the natural killer (NK) cells, B-cells, T-cells, monocytes, and dendritic cells (*Figure 5g* and *Figure 5—figure supplement 4*). Both T cell types showed decreased LINE expression, while the noncytotoxic cluster, TC1, showed slightly decreased LTR expression, and the expanding cytotoxic T cells, TC2, showed decreased SINE expression. On the other hand, the cytotoxic T cells also demonstrated an increased level of inflammatory cytokines and inflammaging gene signature scores. In addition, the NK cells displayed higher senescence and SASP scores in

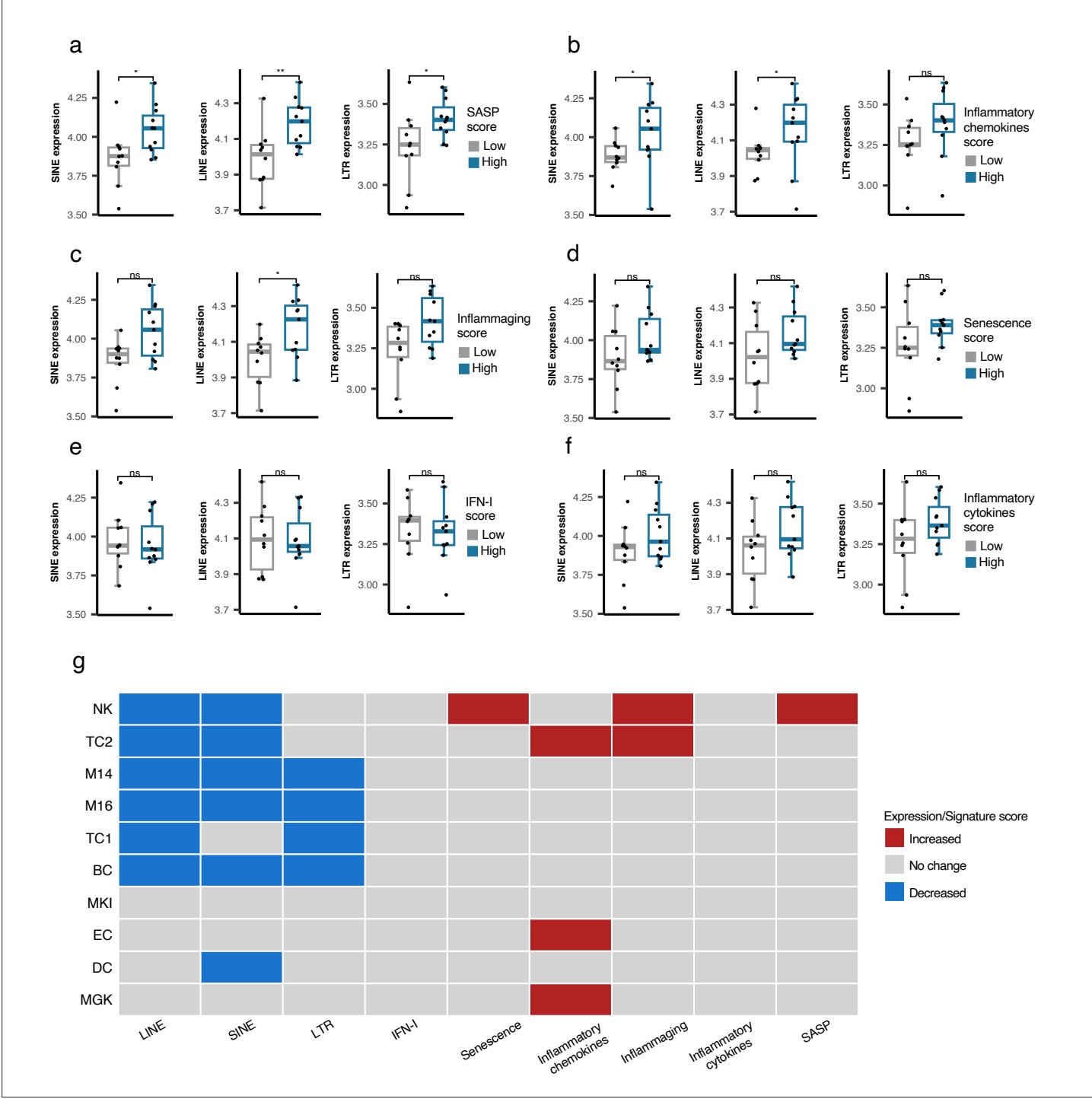

**Figure 5.** Cell type-specific analysis of retrotransposon (RTE) expression vs biological age-related (BAR) gene signature scores in two peripheral blood mononuclear cell (PBMC) scRNA-seq cohorts. (**a**-**f**), Unique increased expression of RTEs in the Plasma B cells of the samples with high biological age-related (BAR) gene signature scores indicates the potential role of plasma B cells in aging. *p≤0.05, **p≤0.01, ns: not significant, Wilcoxon test. n=21. (**g**) Decreased expressions of RTE classes and increased BAR gene signature scores in multiple annotated cell types obtained from the PBMCs of supercentenarians compared to ordinary elderlies as control. Supercentenarians, n=7; control, n=5, age 50–80. Wilcoxon test was applied to identify the significant changes. NK, Natural killer cell; BC, B-cell; TC1, T-cell 1; TC2, T-cell 2; M14, CD14 + monocyte; M16, CD16 + monocyte; EC, Erythrocytes; MKI, MKI67 + proliferating cell; DC, Dendritic cell; MGK, Megakaryocyte.

The online version of this article includes the following figure supplement(s) for figure 5:

*Figure 5 continued on next page*

*Figure 5 continued*

**Figure supplement 1.** Cell type-specific SINE expression versus low and high senescence associated secretory phenotype (SASP) score groups among 25 cell types in peripheral blood mononuclear cell (PBMC).

**Figure supplement 2.** Cell type-specific LINE expression versus low and high senescence associated secretory phenotype (SASP) score groups among 25 cell types in peripheral blood mononuclear cell (PBMC).

**Figure supplement 3.** Cell type-specific long terminal repeats (LTR) expression versus low and high senescence associated secretory phenotype (SASP) score groups among 25 cell types in peripheral blood mononuclear cell (PBMC).

**Figure supplement 4.** Comparison of cell type-specific retrotransposon (RTE) expression in supercentenarians versus normal aged cases.

**Figure supplement 5.** Comparison of cell type-specific Senescence, inflammaging, and senescence associated secretory phenotype (SASP) gene signature scores in supercentenarians versus normal aged cases.

**Figure supplement 6.** Comparison of cell type-specific Inflammatory cytokines, Inflammatory chemokines, and type I interferon (IFN-I) gene signature scores in supercentenarians versus normal aged cases.

the supercentenarians while demonstrating decreased LINE and SINE expressions (*Figure 5g* and *Figure 5—figure supplements 5–6*). Overall, this result reveals inflammatory activity in cytotoxic T cells and increased senescence in the NK cells that are not associated with the activation of RTEs in supercentenarians. By contrast, the expression of RTEs is significantly reduced in most of the immune cell types in supercentenarians.

## Discussion

The role of RTEs in aging has been studied extensively but not in a systematic manner with large, non-cancerous human cohorts across RTE categories before this study. Current understanding of the relationship between RTEs and aging are largely conducted in vitro or in model organisms *Wood et al., 2016*; *Simon et al., 2019*; *LaRocca et al., 2020* that might not apply to humans due to intraspecies differences (*Venuto and Bourque, 2018*). Moreover, the link of RTEs to aging in non-cancerous human cohorts has not been studied systematically in large cohorts.

We combined bulk and single-cell transcriptomic data to examine the relationship between RTE expression versus chronological and biological aging, demonstrating that chronological aging is not significantly linked with RTE reactivation. In this process, we discovered that LTR and LINE expressions are positively correlated with the BAR gene signature scores. However, in contrast, the SINE expression demonstrates an inverse correlation with BAR scores in the PBMC samples and no significant relationship in the WB. We suspect the absence of a positive correlation between BAR scores and SINE expression might be linked to the same discrepancy observed in the GSVA analysis, in which DNA repair pathways are upregulated in the groups of cases with high SINE and Alu expressions. This result suggests that reactivation of SINE, particularly Alu, might lead to DNA damage, a hallmark of aging. Although SINEs compose up to 13% of the human genome, and Alu elements make up the largest family of human mobile elements, Alu is not studied as extensively as L1 from the LINE class (*Bourque et al., 2018*). Therefore, this novel discovery requires further study to confirm the connection and elucidate the mechanism by which DNA repair might be linked to the derepression of SINE and Alu expressions. In contrast, the groups of cases with high LINE and LTR expressions have upregulation of the pathways related to the inflammatory response, which is in line with the previous findings from human and model organisms (*Gorbunova et al., 2021*; *LaRocca et al., 2020*).

The mechanisms of aging and age-related diseases (ARDs) converge on inflammaging (*Franceschi et al., 2018*). Previous studies have shown RTE's pathogenetic role in ARDs including neurodegenerative diseases through elevation of DNA damage and genome instability (*Kaneko et al., 2011*; *Andrenacci et al., 2020*; *Fukuda et al., 2021*). SINE expression inversely correlated with the BAR signature scores in the PBMCs of osteoarthritis patients, while a positive correlation is observed in monocytes. Moreover, a discrepant upregulation of DNA repair pathways is seen in the osteoarthritis cohorts expressing higher level L2, ERV1, Alu, and MIR. With osteoarthritis being a typical ARD, our analysis provides insight into the relationship between aging and RTEs in an ARD state in comparison with that of non-ARD individuals.

The relationship between DNA methylation and expression of RTEs with age was also investigated in our study. The methylation levels of all three RTE classes decrease with chronological age in our

analysis, and the RTE expression is in inverse correlation with the methylation level in LINE and LTR but not in SINE. RTE sequences are repressed via DNA methylation and heterochromatinization (*Deniz et al., 2019*). In the aging process, surveillance of transcriptional regulators such as SIRT6 wanes, leading to hypomethylation and heterochromatin reduction (*Van Meter et al., 2014*). L1 has been identified to be repressed through DNA methylation (*Woodcock et al., 1997*). However, SINE has been suggested to be predominantly regulated by histone modification, and loss of DNA methylation has little effect on the transcription of SINEs (*Varshney et al., 2015*), as reflected in our analysis.

Expression microarray has been successfully employed to study the expression of RTEs (*Balaj et al., 2011*; *Reichmann et al., 2012*; *Wang-Johanning et al., 2012*). However, the probe design of microarrays does not accommodate specifically to the repetitive and interspersed nature of RTEs (*Lanciano and Cristofari, 2020*). To circumvent this issue, our analysis was conducted on the class and family levels but not subfamily levels of RTEs to retain enough probes (*Figure 1—figure supplement 1*) providing sufficient power for calculating the expression scores. We also supported our findings with scRNA-seq, particularly when investigating the association between RTE expression and chronological age, to circumvent the potential constraints posed by using microarray data. However, we are aware of the limitations imposed by using microarray in this study, particularly the low number of RTE probes in the expression microarray data. Although we included one recently published RNA-seq dataset to validate our microarray result, our study can be enriched with the advent of large RNA-seq cohorts for aging studies in the future.

Our exploration of multiple cell types based on a published PBMC scRNA-seq cohort unveiled plasma cells as the only cell type for which the expression of RTEs correlates with BAR signature scores. This matches a recent finding showing a late senescent-like phenotype marked by the accumulation of TEs in plasma cells during pre-malignant stages (*Borges et al., 2023*). Lastly, we explored a supercentenarian PBMC scRNA-seq cohort as a model for healthy aging and identified that RTE expression is generally decreased in supercentenarians. However, the BAR signature scores, particularly senescence and SASP scores, increased in the NK cells of supercentenarians compared to the normal-aged group, suggesting that immunosenescence is an important factor in aging.

# Materials and methods
## Illumina HT12-v4 probes
The probe lists of Illumina Human HT-12 V3 (29431 probes) and V4 (33963 probes) share 29311 probes in common, which is 99.6% of the list of V3. Therefore, we proceeded with HT-12 V4 probes throughout the analysis to cover all studies that used either V3 or V4. To identify the Illumina probes covering RTE regions, we first selected the Illumina Human HT-12 v4 probes covering intergenic or intronic regions, obtaining a list of 924 unique probes. We then overlapped these probe locations with RepeatMasker (*Smit et al., 2013*) regions to acquire the probes covering RTE regions. In total, we could find 232 unique probes covering RTE regions.

## Generating gene signature scores using *singscore*
For each of the curated gene sets, instead of looking at individual gene expressions, we used *singscore* (version 1.20.0) (*Foroutan et al., 2018)*, a method that scores gene signatures in single samples using rank-based statistics on their gene expression profiles, to calculate the gene set enrichment scores. We first compiled the microarray expression matrix for average expression values from RPM values using *limma* (*Ritchie et al., 2015*) package in R, then we used the *rankGenes* function from the *singscore* package to rank each gene sample-wise. Eventually, the *multi Score* function was used to calculate signature scores for all six gene sets at once.

## Statistical analyses
All statistical analyses were performed in R (version 4.3.0). Pearson correlation test was used to determine the r and p values in correlation analyses. For differential expression analysis, the Wilcoxon test is used to determine the significance. The threshold to determine significance is set at *p-value* < 0.05.

## Gene set variation analysis (GSVA)

GSVA was performed using the *GSVA* package (version 1.48.3) *Hänzelmann et al., 2013* in R to compare the groups of low and high RTE expression. The gene lists were retrieved from the Molecular Signatures Database (MSigDB) *Subramanian et al., 2005* by searching for 'inflammatory' and 'dna_repair' as keywords. We further removed pathways that ended with '_DN' (as for downregulated) or '_UP' (as for upregulated) to reduce repeat and confusion. Three repetitive gene sets were removed from our analysis (GOBP_POSITIVE_REGULATION_OF_ACUTE_INFLAMMATORY_RESPONSE, GOBP_POSITIVE_REGULATION_OF_CYTOKINE_PRODUCTION_INVOLVED_IN_INFLAMMATORY_RESPONSE, GOBP_POSITIVE_REGULATION_OF_ACUTE_INFLAMMATORY_RESPONSE_TO_ANTIGENIC_STIMULUS). The filtering process generated 50 gene sets for inflammatory response and 21 gene sets for DNA repair. To detect any alterations in gene expression, the log fold change threshold was kept low at $|logFC| > 0.1$, and a significance threshold of p-value $< 0.05$ was set.

## ScRNA-seq analysis workflow for RTEs

Two single-cell transcriptomic datasets of healthy human cohorts were adopted in this study (*Supplementary file 6*). To obtain the RTE expression in single-cell sequencing (scRNA-seq) data, the scRNA-seq bam files were processed through the scTE (*He et al., 2021*) pipeline. More specifically, for each cell in each sample, the read counts for different classes and families of RTEs were generated using the scTE method. Subsequently, in each sample, we calculated the cumulative read counts for each RTE class per annotated cell type by adding the number of reads belonging to all the cells of each cell type. This resulted in pseudo-bulk read counts of RTE classes for different cell types in each sample.

In parallel, we also calculated the pseudo-bulk read counts for all genes per cell type in each sample using the genic read counts embedded in Seurat (*Satija et al., 2015*) objects collected for the two scRNA-seq datasets that we employed in our study (*Supplementary file 6*). Concatenating pseudo-bulk RTE and gene read counts, we performed reads per million (RPM) normalization per cell type per sample. For each cell type, the denominator for RPM normalization was the cumulative counts of genes and RTE classes for that cell type.

To calculate the age-associated gene signature scores, we converted the scRNA-seq read counts to bulk RNA-seq read counts by summing up the number of reads of all cells of the same type for each gene. This resulted in a bulk RNA-seq gene count matrix from which we calculated each sample's RPM values per gene. Next, the RPM matrix was provided to *singscore* to calculate gene signature scores. Lastly, we divided the samples into high vs low gene signature scores by using the median as the cut-off.

## Bulk RNA-seq analysis for RTEs

The raw RNA-seq reads were mapped to RepeatMasker to extract the reads covering the RTE regions, and then aggregated and normalised for each class and family of RTEs to generate RPM scores.

# Acknowledgements

This work was supported by the Matt Wilson Scholarship from the Faculty of Life Sciences & Medicine, King's College London. This study was supported with funding from Bristol Myers Squibb (BMS) company during this project. The scRNA-seq raw and processed data of healthy aging populations and the supercentenarians were provided by Maxim N Artyomov from Washington University School of Medicine in St. Louis and Piero Carninci from RIKEN Center for Integrative Medical Sciences, Yokohama, respectively. We thank Dr. D Mager for the critical reading of the manuscript.

# Additional information

### Competing interests

Mohammad M Karimi: Reviewing editor, *eLife*. The other authors declare that no competing interests exist.

## Funding

| Funder | Grant reference number | Author |
|---|---|---|
| King's College London | Matt Wilson Scholarship | Yi-Ting Tsai |

The funders had no role in study design, data collection and interpretation, or the decision to submit the work for publication.

## Author contributions

Yi-Ting Tsai, Conceptualization, Data curation, Writing – original draft, Writing – review and editing; Nogayhan Seymen, Conceptualization, Data curation, Software, Formal analysis, Visualization, Methodology, Writing – review and editing; I Richard Thompson, Xinchen Zou, Formal analysis, Methodology; Warisha Mumtaz, Sila Gerlevik, Formal analysis; Ghulam J Mufti, Resources, Funding acquisition, Writing – review and editing; Mohammad M Karimi, Conceptualization, Supervision, Funding acquisition, Investigation, Visualization, Methodology, Writing – original draft, Project administration, Writing – review and editing

## Author ORCIDs

Nogayhan Seymen (iD) http://orcid.org/0000-0002-2379-5542
Sila Gerlevik (iD) https://orcid.org/0000-0001-6617-1310
Mohammad M Karimi (iD) https://orcid.org/0000-0001-5017-1252

Reviewer #1 (Public review): https://doi.org/10.7554/eLife.96575.4.sa1
Reviewer #2 (Public review): https://doi.org/10.7554/eLife.96575.4.sa2
Author response https://doi.org/10.7554/eLife.96575.4.sa3

# Additional files

## Supplementary files

• Supplementary file 1. Microarray datasets.

• Supplementary file 2. Number of expression probes mapping RTE classes across cohorts.

• Supplementary file 3. Number of expression probes mapping RTE families across cohorts.

• Supplementary file 4. List of RTE-covering probes from Illumina HumanHT-12-v4 Expression assay.

• Supplementary file 5. List of TE-covering probes from the Illumina Infinium 450 k array.

• Supplementary file 6. scRNA-seq data.

• Supplementary file 7. Adopted gene lists to describe age-related events.

• Supplementary file 8. Genes missing from the biological age-related (BAR) gene lists in the GTP dataset.

• MDAR checklist

## Data availability

The current manuscript is a computational study, so no data have been generated for this manuscript. Scripts and data used in this study are available on Github (https://github.com/Karimi-Lab/TE_aging_manuscript; copy archived at *Seymen and Thompson, 2024*).

The following previously published datasets were used:

| Author(s) | Year | Dataset title | Dataset URL | Database and Identifier |
|---|---|---|---|---|
| Reynolds LM, Taylor JR, Ding J, Lohman K | 2014 | Age-related variations in the methylome associated with gene expression in human monocytes and T cells | https://www.ncbi.nlm.nih.gov/geo/query/acc.cgi?acc=GSE56045 | NCBI Gene Expression Omnibus, GSE56045 |

*Continued on next page*

*Continued*

| Author(s) | Year | Dataset title | Dataset URL | Database and Identifier |
|---|---|---|---|---|
| Mehta D | 2014 | Transcriptional landscape of aging in humans | https://www.ncbi.nlm.nih.gov/geo/query/acc.cgi?acc=GSE58137 | NCBI Gene Expression Omnibus, GSE58137 |
| Ramos YF, Bos SD, Lakenberg N, Böhringer S | 2014 | Genes expressed in blood link osteoarthritis with apoptotic pathways | https://www.ncbi.nlm.nih.gov/geo/query/acc.cgi?acc=GSE48556 | NCBI Gene Expression Omnibus, GSE48556 |
| Wang et al. | 2018 | Epigenetic influences on aging: a longitudinal genome-wide methylation study in old Swedish twins | https://www.ebi.ac.uk/biostudies/arrayexpress/studies/E-MTAB-7309 | Bio Studies, E-MTAB-7309 |
| McRae AF, Powell JE, Henders AK, Bowdler L | 2014 | Contribution of genetic variation to transgenerational inheritance of DNA methylation | https://www.ncbi.nlm.nih.gov/geo/query/acc.cgi?acc=GSE56105 | NCBI Gene Expression Omnibus, GSE56105 |
| Hannum G, Guinney J, Zhao L, Zhang L | 2012 | Genome-wide methylation profiles reveal quantitative views of human aging rates | https://www.ncbi.nlm.nih.gov/geo/query/acc.cgi?acc=GSE40279 | NCBI Gene Expression Omnibus, GSE40279 |
| Mogilenko DA | 2021 | Comprehensive Profiling of an Aging Immune System Reveals Clonal GZMK(+) CD8(+) T Cells as Conserved Hallmark of Inflammaging | https://www.ncbi.nlm.nih.gov/geo/query/acc.cgi?acc=GSE155006 | NCBI Gene Expression Omnibus, GSE155006 |
| Hashimoto K | 2019 | Single-cell transcriptomics reveals expansion of cytotoxic CD4 T cells in supercentenarians | https://gerg.gsc.riken.jp/SC2018/ | RIKEN, SC2018 |

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
