## [Editor Report · eLife Assessment]

The study by Tsai et al. employed multi-omics approaches, including transcriptomic, methylomic, and single-cell RNA-seq, and provided a **solid** and comprehensive analysis of the correlation between retrotransposable element (RTE) expression and biological aging in human blood. Their findings highlight the differential roles of RTE families, providing **valuable** insights for understanding the mechanisms of human aging.

---

## [Referee Report · Reviewer #1 (Public review)]

Tsai and Seymen et al. investigate associations between RTE expression and methylation and age and inflammation, using multiple public datasets. The text of the manuscript has been polished and the phrasing of several findings has been made clearer and more precise. The authors also provided ample discussion to the prior reviewer comments in their rebuttal, including new analyses.

---

## [Referee Report · Reviewer #2 (Public review)]

Summary:

Yi-Ting Tsai and colleagues conducted a systematic analysis of the correlation between the expression of retrotransposable elements (RTEs) and aging, using publicly available transcriptional and methylome microarray datasets of blood cells from large human cohorts, as well as single-cell transcriptomics. Although DNA hypomethylation was associated with chronological age across all RTE biotypes, the authors did not find a correlation between the levels of RTE expression and chronological age. However, expression levels of LINEs and LTRs positively correlated with DNA demethylation, and inflammatory and senescence gene signatures, indicative of "biological age". Gene set variation analysis showed that the inflammatory response is enriched in the samples expressing high levels of LINEs and LTRs. In summary, the study demonstrates that RTE expression correlates with "biological" rather than "chronological" aging.

Strengths:

The question the authors address is both relevant and important to the fields of aging and transposon biology.

Comments on latest version:

The authors introduced the analysis of RNA-seq data, addressing the key concerns raised by Reviewer #1 and myself. They also adopted more explicit terminology in their latest version, reducing ambiguity. The RNA-seq analysis demonstrating that the expression of different transposon groups is not associated with chronological aging is convincing, though, in my opinion, it still lacks granularity.

I have two minor points:

(1) Previously, I have mentioned the following:

"The authors pool signals from RTEs by class or family, despite the fact that these groups include subfamilies and members with very different properties and harmful potentials. For example, while older subfamilies might be expressed through readthrough transcription, certain members of younger groups could be autonomously reactivated and cause inflammation... The aggregation of signals from different RTE biotypes may obscure potential reactivation of smaller groups or specific subfamilies."

The authors responded that they would lose statistical power by studying RTE subfamilies with limited microarray probes, which is a fair point. However, the suggested analysis could have been conducted using the RNA-seq data they explored in the second round of revision. Choosing not to leverage RNA-seq to increase the granularity of their analysis is a matter of choice. In my opinion, however, the authors could have acknowledged in the discussion that some smaller yet potentially influential RTE species may be masked by their global approach.

(2) Previously, I mentioned that 10x scRNA-seq is not ideal for analysing RTEs and requested a classical UMAP plot to visualize RTE expression across cell populations. The authors argued that they could only achieve sufficient statistical power by quantifying RTE classes through cumulative read counts for each cell type, which I accept. However, they divided cells into "high" and "low" BAR gene signature groups. I am surprised that the comparison of BAR signature expression between these groups was not presented using standard visualization methods commonly applied in scRNA-seq data analysis.

---

## [Author Response]

The following is the authors’ response to the current reviews.

**Reviewer #2 (Public Review):**
The authors responded that they would lose statistical power by studying RTE subfamilies with limited microarray probes, which is a fair point. However, the suggested analysis could have been conducted using the RNA-seq data they explored in the second round of revision. Choosing not to leverage RNA-seq to increase the granularity of their analysis is a matter of choice. In my opinion, however, the authors could have acknowledged in the discussion that some smaller yet potentially influential RTE species may be masked by their global approach."

We will add one sentence addressing this in the Version of Record.

The following is the authors’ response to the original reviews.

We thank Reviewer #1 for their constructive comments.

**Public Reviews:**

**Reviewer #1 (Public Review):**
Tsai and Seymen et al. investigate associations between RTE expression and methylation and age and inflammation, using multiple public datasets. Compared to the previous round of review, the text of the manuscript has been polished and the phrasing of several findings has been made clearer and more precise. The authors also provided ample discussion to the prior reviewer comments in their rebuttal, including new analyses. All these changes are in the correct direction, however, I believe that part of the content of the rebuttal should be incorporated in the main text, for reasons that I will outline below.Both reviewers found the reliance on microarray expression data to detract from the study. The authors argued that their choices are supported by existing publications which performed a similar quantification of TE expression using microarray data. It could still be argued that (as far as I can tell) Reichmann et al. used a substantially larger number of probes than this study, as a consequence of starting from different arrays, however, this is a minor point which the authors do not need to address. It is still undeniable that including the validation with RNA-seq data performed in the rebuttal would strengthen the manuscript. I especially believe that many readers would want to see this analysis be prominent in the manuscript, considering that both reviewers independently converged on the issue with microarray expression data. Personally, I would have included an RNA-seq dataset next to the microarray data in the main figures, however, I understand that this would require considerable restructuring and that placing RNAseq data besides array data might be misleading. Instead, I would ask that the authors include their rebuttal figures R1 and R2 as supplementary figures.I would suggest introducing a new paragraph, between the section dedicated to expression data and the one dedicated to DNA methylation, mentioning the issues with microarray data (Some of which were mentioned by the reviewers and other which were mentioned by the authors in the discussion and introduction) to then introduce the validation with RNA-seq data.

We appreciate the reviewer’s understanding and detailed feedback. As suggested, Author response images 1 and 2 were added as supplementary figures to the manuscript, and one paragraph was added to the section investigating the correlation between RTE expression and chronological age. We have also added new descriptions to the introduction, discussion, and BAR analysis sections.

Author response image 3 is also a good addition and should be expanded to include the GTP and MESA study and possibly mentioned in the paragraph titled "RTE expression positively correlates with BAR gene signature scores except for SINEs."

We have updated Author response image 3 (now Author response image 1) to include GTP and MESA cohorts in the analysis. As shown in Author response image 1, except IFN-I and senescence scores on the MESA cohort that positively correlate with chronological ageing, the rest of the gene signatures display no positive correlation with chronological ageing.

Author response image 1 was originally created to separate the effect of chronological age and RTE expression on BAR gene signature scores. As it was meant to discriminate between BAR and chronological age, it doesn't provide additional information regarding the positive correlation between RTE expression and BAR gene signature that was not already present in the manuscript. Therefore, we did not add it to the manuscript.

**Author response image 1. sa3fig1:** Generalized linear models (GLM) analysis (BAR gene signature scores ~ RTE expression +chronological age). For each RTE family, we separately performed GLM. Age (RTE family) indicates the chronological age when used in the design formula for that specific RTE family.

"In this study, we did not compare MESA with GTP etc. We have analysed each dataset separately based on the available data for that dataset. Therefore, sacrificing one analysis because of the lack of information from the other does not make sense. We would do that if we were after comparing different datasets. Moreover, the datasets are not comparable because they were collected from different types of blood samples."Indeed, the datasets are not compared directly, but the associations between age, BER and TE expression for each dataset are plotted and discussed right next to each other. It is therefore natural to wonder if the differences between datasets are due to differences in the type of blood sample or if they are a consequence of the different probe sets. Using a common set of probes would help answer that question.

We understand that the reviewer is proposing a method to eliminate the possible causes of differences across datasets. However, incorporating such change would compromise the statistic power of MESA and GARP cohorts and also change our analysis structurally and digress from our main focus. Hence, we disagree to use the identical set of probes for all three cohorts.